# Robust learning from noisy, incomplete, high-dimensional experimental data via physically constrained symbolic regression

Patrick A. K. Reinbold[1], Logan M. Kageorge[1], Michael F. Schatz [1] & Roman O. Grigoriev [1✉]

Machine learning offers an intriguing alternative to first-principle analysis for discovering new physics from experimental data. However, to date, purely data-driven methods have only proven successful in uncovering physical laws describing simple, low-dimensional systems with low levels of noise. Here we demonstrate that combining a data-driven methodology with some general physical principles enables discovery of a quantitatively accurate model of a non-equilibrium spatially extended system from high-dimensional data that is both noisy and incomplete. We illustrate this using an experimental weakly turbulent fluid flow where only the velocity field is accessible. We also show that this hybrid approach allows reconstruction of the inaccessible variables – the pressure and forcing field driving the flow.

[1] School of Physics, Georgia Institute of Technology, Atlanta, GA, USA. ✉email: roman.grigoriev@physics.gatech.edu

Revolutionary advances in our ability to collect, store, and process vast amounts of information has unleashed machine learning as a dramatically different approach to scientific discovery[1–3]. Initial efforts have focused on purely data-driven methods to synthesize knowledge in the form of equations. For instance, symbolic regression has been applied successfully to extract both evolution laws expressed as ordinary differential equations[4] and conservation laws in the form of algebraic equations[5] from low-dimensional data with low levels of noise. Unfortunately, to date, purely data-driven approaches have been unable to handle high-dimensional data sets representing complex or spatially extended non-equilibrium phenomena such as cancer, fusion plasmas, earthquakes, weather, or climate change. A key difficulty is that, without appropriate constraints, the high dimensionality of the data makes the model search space far too large for any purely data-driven approach to be tractable.

In principle, machine learning can be used to construct suitable models (e.g., nonlinear partial differential equations (PDEs)) of spatially extended systems[6,7]; however, numerous difficulties arise when using data from the real world. First and foremost, all the variables (or fields) that are necessary to describe the phenomena of interest should be identified; no existing purely data-driven approach can help with this. Second, some of the required variables may not be accessible in a real-world problem; to date, no known machine learning method has been successful in model discovery based on incomplete data. Third, data from real-world problems often involve significant uncertainty due to both random and systematic errors, which, as a consequence, makes accurate evaluation of particular, crucially important model terms infeasible. Finally, unlike the test cases using synthetic data generated by a reference model[6,7], assessing the quality of a model learned from real-world data is not straightforward. The fusion of domain knowledge with data science[8] is essential for addressing these challenges.

Here we present such a hybrid approach that uses appropriate physical constraints (e.g., locality, smoothness, symmetries) to dramatically constrain the search space containing various candidate models. Our approach incorporates three key ingredients: (1) general physical principles used to identify the variables and candidate models, (2) weak formulation of differential equations to reduce noise sensitivity and eliminate dependence on inaccessible variables, and (3) ensemble symbolic regression to identify a parsimonious model that balances accuracy and simplicity. To illustrate, we examine an experimental fluid flow in a thin layer that exhibits complex spatiotemporal behavior when driven by time-independent forcing[9] (see Fig. 1 and the "Methods" section). We show that a quantitative 2D model of this flow can be discovered using experimental measurements of the horizontal components of the velocity field $\mathbf{u}(\mathbf{x}, t)$. Furthermore, using this model, all latent fields (here pressure and forcing) can also be reconstructed.

We start by describing the three key components of the hybrid approach to model discovery. Additional details are provided in the "Methods" section. The first two steps of model discovery are to identify a set of variables (fields) required to describe the data and construct a sufficiently broad library of candidate models that will later be narrowed down to obtain a parsimonious description. In practice, these two steps may be hard, or even impossible, to separate and, for systems of high dimensionality, require additional considerations based on domain knowledge. For the system considered here, the general physical assumptions of causality, locality, and smoothness can be used to write the model in the form of Volterra series[10]. Each term $\mathbf{F}_n$ of the series involves a product of the velocity field $\mathbf{u}$, latent fields, and/or their partial derivatives. Since we are dealing with a fluid flow, we can rely on the more specific domain knowledge recognizing the fluid flow is driven by external and internal stresses. Hence, the evolution of the velocity field should depend on body forces $\mathbf{f}$ and pressure $p$, which are the latent fields here:

$$\partial_t \mathbf{u} = \sum_n c_n \mathbf{F}_n[\mathbf{u}, p, \mathbf{f}, \nabla \mathbf{u}, \nabla p, \nabla \mathbf{f}, \dots \ ]. \tag{1}$$

The library of candidate models can be further constrained by using another general physical concept of Euclidean symmetry which reflects the uniformity and isotropy of the fluid layer. Truncating the sum at a sufficiently low order in the fields and derivatives yields[11]

$$\begin{aligned}\partial_t \mathbf{u} = \ & c_1(\mathbf{u} \cdot \nabla)\mathbf{u} + c_2 \nabla^2 \mathbf{u} + c_3 \mathbf{u} + c_4 u^2 \mathbf{u} + c_5 \omega^2 \mathbf{u} \\ & + c_6(\nabla \cdot \mathbf{u})\mathbf{u} + c_7(\nabla \cdot \mathbf{u})^2 \mathbf{u} - \rho^{-1}\nabla p + \rho^{-1}\mathbf{f},\end{aligned} \tag{2}$$

where $\omega = \hat{z} \cdot (\nabla \times \mathbf{u})$ is the vorticity and $u^2 = \mathbf{u} \cdot \mathbf{u}$. Isotropy constrains the functional form of the library terms, each of which transforms as a vector, while uniformity implies that the unknown coefficients are constants, i.e., independent of position and time. Note that, without loss of generality, the coefficients of the last two terms can be set to $\pm\rho^{-1}$, where $\rho$ is an arbitrary constant with the units of mass density; this simply amounts to fixing the units (and sign) of the pressure and forcing fields.

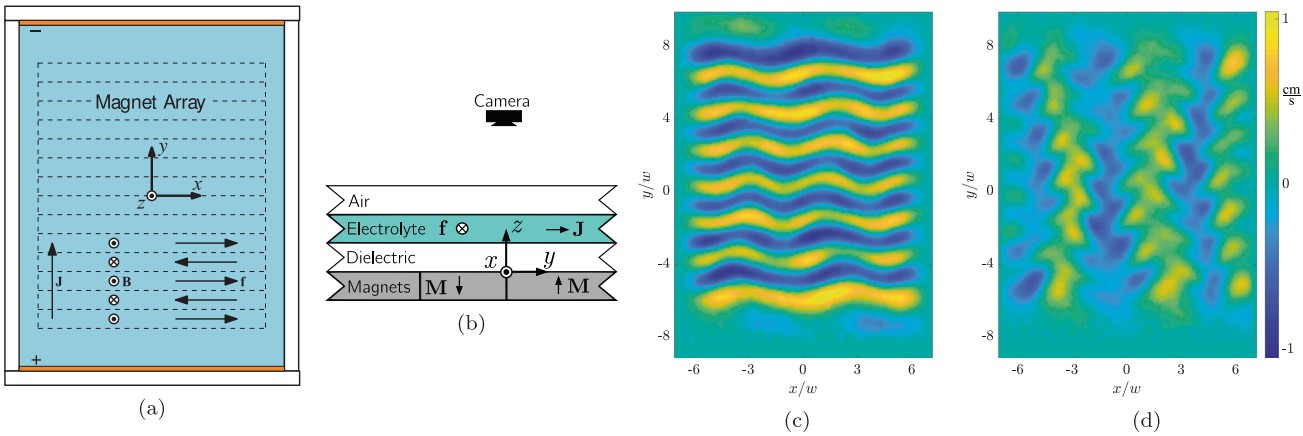

**Fig. 1 Experimental setup and sample data.** Schematic top (**a**) and side (**b**) views are shown for laboratory studies of weak turbulence in a thin electrolyte layer inside a rectangular container. Flow is driven by Lorentz forcing **f**, which arises by applying a current density **J** in the presence of a magnetic field **B** from a permanent magnet array (dashed lines). The heatmaps illustrate snapshots of measured velocity fields in the $x$- (**c**) and $y$- (**d**) directions at Reynolds number Re = 22.17, when the flow is weakly turbulent.

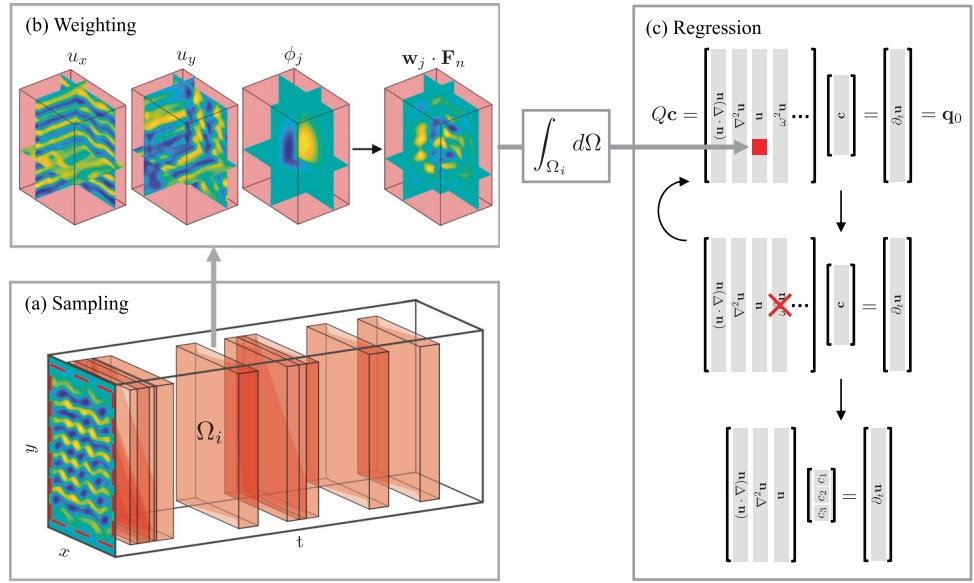

**Fig. 2 Symbolic regression algorithm based on weak formulation. a** Integration domains, shown as red boxes, are randomly sampled throughout the 2D space-1D time data set. **b** For each integration domain $\Omega_i$, the data $\mathbf{u} = u_x \hat{x} + u_y \hat{y}$ and the weights $\mathbf{w}_j = \nabla \times [\phi_j \hat{z}]$ are used to evaluate the scalar product $\langle \mathbf{w}_j \cdot \mathbf{F}_n \rangle$, as discussed in the "Methods" section. The result determines the matrix element $Q_{kn}$ (for $n \neq 0$) or the $k$-th element of $\mathbf{q}_0$ (for $n = 0$), where the composite index $k$ runs over all integration domains $i$ and weights $j$. The columns are labeled using the corresponding terms in the model instead of the index $n$ to make the relation with the linear system (6) more transparent. **c** A sparse solution to the system is then found via sequential thresholding, where one (or more) columns are removed from the matrix $Q$ (and the model) at each iteration, until a parsimonious model balancing accuracy with simplicity is identified (bottom of (**c**)).

While the forcing in this particular experiment is time-independent, the pressure varies in time and so requires its own model. A corresponding library of candidate models can constructed in a similar way which, after truncation to lowest-order terms, yields

$$\partial_t p = c_8 \nabla \cdot \mathbf{u} + c_9 \nabla \cdot \mathbf{f} + c_{10} p. \tag{3}$$

Here each term transforms as a scalar, and $c_8$, $c_9$, and $c_{10}$ are additional unknown constants. We can further constrain both libraries using the experimental observation that, to high accuracy, the velocity field is divergence-free, which corresponds to setting $c_6 = c_7 = 0$ in Eq. (2) and $c_8 \to \infty$ in Eq. (3).

The need for including in the model the dependence on the pressure and forcing fields could be discovered from data directly without relying on the knowledge of fluid dynamics. We can rewrite Eq. (2) in the form

$$\rho \mathbf{s} = -\nabla p + \mathbf{f}, \tag{4}$$

where $\mathbf{s}$ represents the sum of all the terms that depend only on $\mathbf{u}$ and its partial derivatives. In general, we would find $\mathbf{s} \neq 0$ for any choice of the coefficients. Helmholtz decomposition requires $\mathbf{s} = \nabla \phi + \nabla \times \mathbf{A}$, where $\phi$ and $\mathbf{A}$ are the scalar and vector potentials. Hence two additional fields, one scalar and one vector, are required to satisfy Eq. (4): $p = -\rho \phi$ and $\mathbf{f} = \rho \nabla \times \mathbf{A}$.

Although symbolic regression could be performed using the strong form of the model, e.g., by directly evaluating each term in Eq. (2) at different spatiotemporal locations, this presents two problems. The most obvious one is that we cannot evaluate the terms involving latent fields. Pressure could, in principle, be computed by taking the divergence of Eq. (2) and solving the resulting pressure-Poisson equation, if the forcing $\mathbf{f}$ were known or at least divergence-free. In our case, this is not an option, since $\mathbf{f}$ satisfies neither condition. Furthermore, taking a derivative greatly amplifies the noise present in the data, whether this is done using finite differences[6,12], polynomial interpolation[11], or spectral methods[13,14]. Instead, we use a weak form of the model to address both noise sensitivity and the dependence on latent variables. This approach

was originally introduced in the context of ordinary differential equations[15,16]. In the context of PDE models, it was shown to be as general as prior approaches based on the strong form[6,7] and superior in terms of both its flexibility and robustness[17,18].

Let us choose a set of spatiotemporal domains $\Omega_i$ and weight functions $\mathbf{w}_j$ (see the "Methods" section and Fig. 2) and define

$$\langle \mathbf{w}_j, \mathbf{F}_n \rangle_i = \int_{\Omega_i} \mathbf{w}_j \cdot \mathbf{F}_n d\Omega, \tag{5}$$

where $d\Omega = dx\, dy\, dt$ and $n = 0$ corresponds to the term $\partial_t \mathbf{u}$. Evaluating the integrals in equation (5) for different $i$ and $j$ and stacking the results to form vectors $\mathbf{q}_n$, we arrive at a linear system of equations for the unknown coefficients

$$Q\mathbf{c} = \mathbf{q}_0, \tag{6}$$

where $\mathbf{c} = [c_1, \cdots, c_N]^T$ and $Q = [\mathbf{q}_1 \cdots \mathbf{q}_N]$.

A parsimonious model describing the data can be found by solving an overdetermined system (6) using any standard algorithm such as LASSO[19], ridge regression[20], sequentially thresholded least squares[21], or various information-theoretic criteria[22]. Here we adopt the computationally efficient iterative procedure introduced in ref. [18], which is an adaptation of the latter algorithm. At each iteration, Eq. (6) is solved to find parameters $c_1$ through $c_N$. Then, the magnitude of each term is computed. If it is below some threshold, say $\|c_n \mathbf{q}_n\| < \varepsilon \|\mathbf{q}_0\|$ for a given choice of $\varepsilon$, the corresponding term is removed from the library by setting $c_n = 0$ and the column $\mathbf{q}_n$ is removed from the matrix $Q$. The process is then repeated until all remaining terms have a magnitude that is above the threshold.

How well a model describes a particular data set can be quantified in terms of the relative residual

$$\eta = \frac{\| Q\mathbf{c} - \mathbf{q}_0 \|}{\max_n \| c_n \mathbf{q}_n \|}, \tag{7}$$

where we expect $\eta \ll 1$ when all the relevant terms in the model have been identified. The magnitude of $\eta$ however tells us little

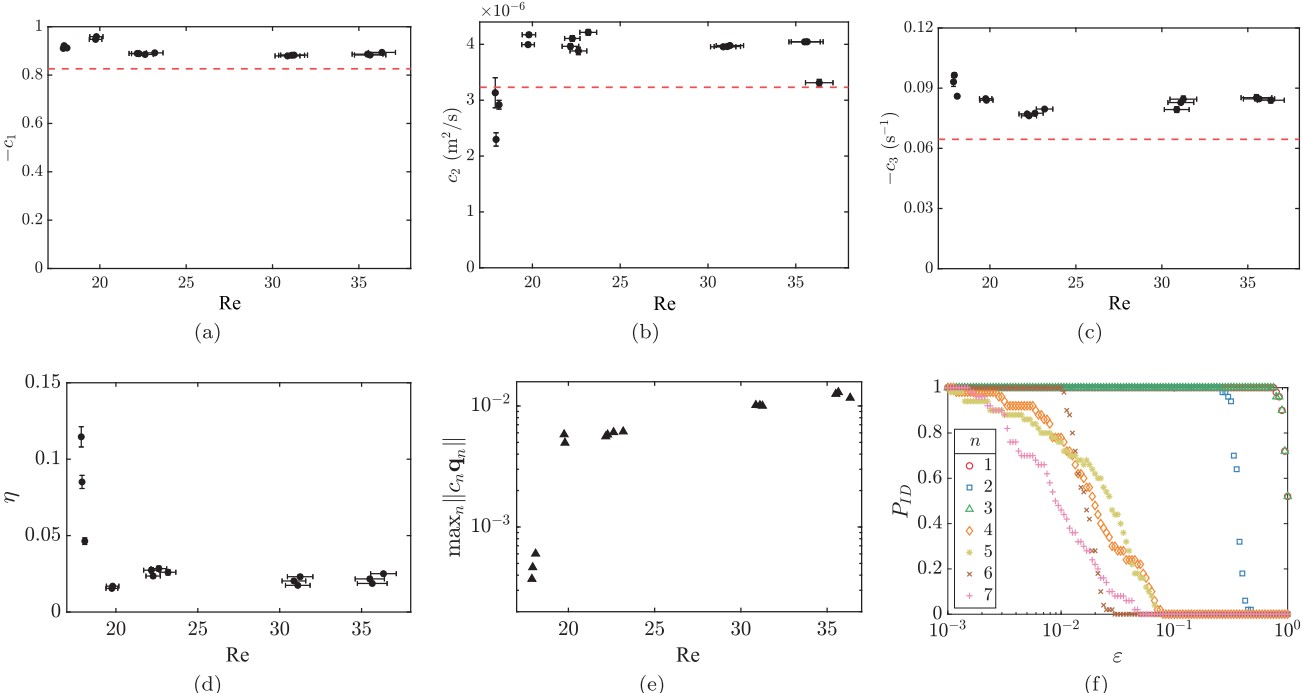

**Fig. 3 Regression results.** Model parameters, shown in (**a–c**) are consistently well estimated from experimental data for a range of Reynolds numbers Re, particularly when the amplitude of flow time dependence is sufficiently large, as illustrated in (**d**) and (**e**). For the results shown, flows in experiments are time-periodic for $Re \lesssim 19$, and weakly turbulent otherwise. In (**a–c**), parameters obtained using ensemble averaging (black dots) are compared with the corresponding value obtained using first-principle analysis (dashed line) performed for time-independent flows at low Re. In (**d**), low values of the residual $\eta$ (Eq. 7) indicate good parameter fits; the relative quality of fit deteriorates in a regime ($Re \lesssim 19$) where flow time dependence is weak and, therefore, the maximum magnitude of terms in Eq. (6) is small (**e**). The terms retained in a parsimonious model depend on a choice of threshold $\varepsilon$; the probability of retaining the term $\mathbf{F}_n$ as a function of $\varepsilon$ shown in (**f**) indicates the model given by Eq. (8) is consistently identified by choosing $0.1 \lesssim \varepsilon \lesssim 0.3$. The vertical error bars in (**a–d**) represent the standard deviation over the ensemble (in most instances they are smaller than the symbol size) and the horizontal error bars represent the variation in Re over the data set.

about the functional form of the model or the magnitude of the respective coefficients. For instance, including a term such as $c_6(\nabla \cdot \mathbf{u})\mathbf{u}$ with an arbitrary coefficient $c_6$ in Eq. (2) does not change $\eta$ for a flow that is incompressible, but does change the model[11]. The robustness of the functional form of the model and the accuracy with which the coefficients $c_n$ are determined can both be quantified by performing symbolic regression for an ensemble of different samplings of the data (or even different data sets)[18]. Here, each ensemble includes different distributions of integration domains in the temporal direction. The variation in the functional form of the identified model across the ensemble can be used to detect missing or spurious terms, while the standard deviation of the coefficients $c_n$ can be used to quantify their accuracy.

## Results
To test our approach for model discovery, we measured the velocity field components in the plane of the fluid layer and performed symbolic regression for an ensemble of 30 different random distribution of spatiotemporal domains $\Omega_i$. We found that choosing $0.1 \lesssim \varepsilon \lesssim 0.3$ gives the best balance of robustness with accuracy (Fig. 3f). For higher $\varepsilon$, the model does not fit the data accurately, as measured by $\eta$. For lower $\varepsilon$, the functional form of the model acquires a sensitive dependence on the choice of spatiotemporal domains $\Omega_i$, which is a sign of overfitting.

Over the range of Reynolds numbers $17.8 \lesssim Re \lesssim 36$, symbolic regression consistently identified a parsimonious model

$$\partial_t \mathbf{u} = c_1(\mathbf{u} \cdot \nabla)\mathbf{u} + c_2 \nabla^2 \mathbf{u} + c_3 \mathbf{u} - \rho^{-1}\nabla p + \rho^{-1}\mathbf{f}, \qquad (8)$$

with $\eta$ as low as 0.02 (see Fig. 3d). This model allows easy interpretation, since its form is similar to the Navier–Stokes equation which represents momentum balance. The first term on the right-hand side describes advection of momentum. The second and third terms describe momentum flux due to viscosity in the horizontal and vertical direction[9,23], respectively. The fourth and fifth terms also appear in the Navier–Stokes equation and describe (isotropic) internal stresses and external stresses, respectively.

It is worth emphasizing that the form of the 2D model identified by symbolic regression is identical to that derived from the first principles[9,24] under a number of assumptions, including the divergence-free condition on the horizontal components of the velocity. Dropping this assumption produces a more general model[25], which is a special case of the system (2)–(3) with $c_6 \neq 0$, $c_7 = 0$, $c_8 \neq \infty$, and $c_9 = c_{10} = 0$. In both cases, the coefficients $c_1$, $c_2$, and $c_3$ are nonzero and given by explicit expressions in terms of the material parameters and the geometry of the fluid layer[9]. The theoretical values of parameters are compared with the respective values identified by symbolic regression in Fig. 3a–c.

Note that all three parameters identified using experimental data are close, but not identical, to the theoretical values (Fig. 3a–c). This helps explain the discrepancy in the critical Re of the primary instability in this system in experiment and numerics[24]. The original study estimated that a 22% increase in the value of $c_3$ would be required to match the observed value with the model predictions, assuming the other two parameters do not change. The identified values of $c_3$ are about 25% higher than the theoretical value (Fig. 3c), which is consistent with that estimate.

The accuracy with which the parameters of the model are estimated via symbolic regression can be judged based on both their standard deviation for each ensemble and the variation of the mean between different data sets at roughly the same $Re$. The former is much smaller than the latter, and so may underestimate the true uncertainty. Different data sets represent separate experiments, so, conversely, the variation in the mean could also reflect the (small) variation in the conditions of the experiment (e.g., the thickness of the fluid layers). While the difference in the mean values of $c_2$ for the two data sets at $Re \approx 36$, where the flow is weakly turbulent, is probably attributed to just such a variation in the conditions, the much larger variation in the mean of $c_2$ and $c_3$ for the three data sets at Re $\approx 18$ (Fig. 3b, c) is most likely due to a qualitative change in the dynamics.

For $17.8 \lesssim Re \lesssim 19$ the flow becomes time-periodic[24]. The amplitude of the temporal oscillation decreases substantially as $Re$ approaches $Re \approx 17.8$, leading to a corresponding decrease in the magnitude of all the terms (Fig. 3e) and an increase in $\eta$ (Fig. 3d). Indeed, the constraint (12) on the weight functions implies that $\langle \mathbf{F}_n, \mathbf{w}_j \rangle = 0$ for all $n$ for a stationary flow. Hence our particular choice of the weight functions is only suitable for flows that are time-dependent. This is the fundamental reason why the accuracy of the reconstructed model decreases at the low end of the $Re$ range explored here, where the magnitude of the time-dependent component of the velocity field becomes comparable to the measurement error of the particle image velocimetry (PIV). The breakdown of our approach for steady flows is not an inherent problem of symbolic regression but is rather due to the presence of latent variables, mainly the steady forcing which the constraint (12) was aimed to eliminate. One way to get around this limitation is to analyze transient flows relaxing toward the steady state.

Once the parsimonious model has been identified, the latent fields can be determined as well. Using the Helmholtz decomposition in Eq. (4), the pressure $p$ and forcing $\mathbf{f}$ can be computed at each time $t$ represented in the data set, as discussed in the "Methods" section. The movie showing the time evolution of the reconstructed pressure field is included as Supplementary Movie 1.

The electrical current is uniform in the electrolyte layer, hence the forcing field $\mathbf{f} = f(x, y)\hat{x}$ that appears in the 2D model of the fluid flow should correspond to the depth average of the Lorentz force across the electrolyte layer:

$$f(x, y) \propto \int J B_z(x, y, z) \, dz. \qquad (9)$$

The forcing profile reconstructed from the measured flow field is compared with the Lorentz force computed from direct experimental measurement of the magnetic field according to Eq. (9) in Fig. 4, which shows that the two profiles are almost indistinguishable.

## Discussion

As we have demonstrated here, a data-driven approach based on symbolic regression can successfully discover a quantitatively accurate model of a fairly complicated and high-dimensional non-equilibrium system with highly nontrivial dynamics using noisy, incomplete experimental measurements. Unlike artificial neural network models[26,27] that trade-off interpretability for generality, our model has the form of a PDE, which is both straightforward to interpret and allows the latent fields to be easily reconstructed. The discovered model can also be directly compared with other models of the same system constructed using first principles. This comparison suggests that the first-principle models do capture all the relevant physical mechanisms

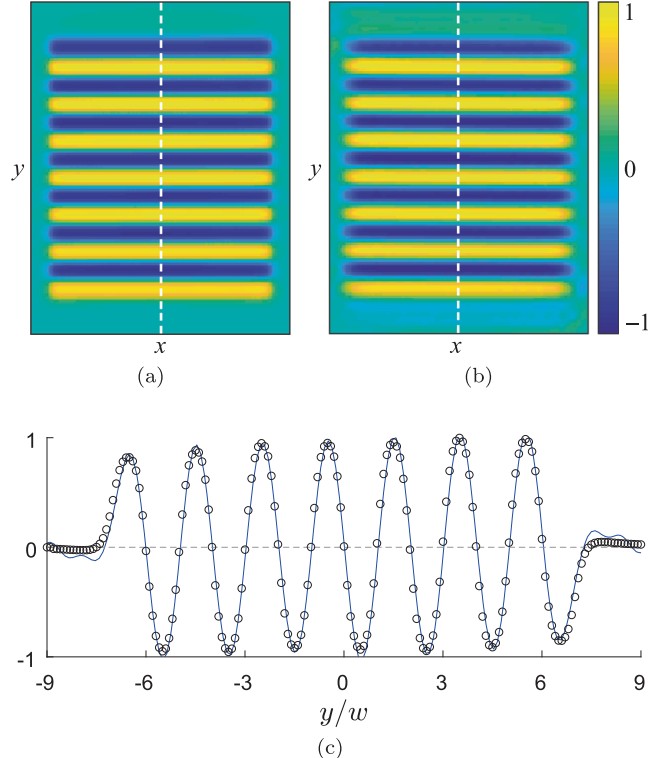

(a)                    (b)

(c)

**Fig. 4 Forcing field.** The $x$-component of the depth-averaged Lorentz force $\mathbf{J} \times \mathbf{B}$ computed using experimental measurement of the magnetic field (**a**) is virtually indistinguishable from the forcing field $\mathbf{f}$ reconstructed using Eq. (8) for $Re = 22.17$ (**b**). In **c**, the reconstructed (blue line) and measured (black circles) forcing profiles, both normalized by their maximum magnitude, are compared along the line $x = 0$ (dashed lines in (**a**) and (**b**)). This normalization which is used in all three panels, also removes the dependence on an arbitrary choice of $\rho$ in Eq. (8).

qualitatively, but fail to describe them quantitatively with sufficient accuracy, indicating that the assumptions used in their derivation require refinement.

Although our results validate the practical utility of data-driven model discovery, they also highlight the need for a hybrid approach that combines a number of general physical constraints —most notably, locality, causality, and spatial symmetries—to generate a library of candidate models with symbolic regression which downselects from this library the parsimonious model that best describes the data. Although purely data-driven approaches such as manifold learning[28] can be used to help with library construction, it is unlikely that this approach remains tractable for high-dimensional systems such as the one considered here. We have also relied on fairly specific domain knowledge to identify the latent fields that are not a part of the data. While in our case, their presence is suggested by the structure of the model, no general approach to identifying latent variables from data has been developed so far.

Domain knowledge also plays an essential role in choosing the weight functions. We used both the functional form of the terms involving the latent variables (e.g., $\nabla p$) and the known properties of the latent fields (e.g., the forcing $\mathbf{f}$ being time-independent) to eliminate the dependence on both $p$ and $\mathbf{f}$ from the regression problem. This would not have been possible without using some domain knowledge, illustrating the limitations of the purely data-driven approach. It should also be mentioned that the dependence on latent fields may not always be eliminated, while still allowing the governing equations to be identified. For instance,

our approach would not succeed without measurement of the velocity field, even if the pressure were known.

The success of any data-driven approach is also heavily dependent on the data used[29]. In particular, for PDE discovery, the data should exhibit variation in all independent coordinates. In the present problem, we find that symbolic regression identifies a sparse model with high accuracy for higher $Re$ where the flow is weakly turbulent and the velocity field varies in time and both spatial coordinates. The same exact approach experiences difficulties at lower $Re$ where the flow becomes (nearly) stationary. Indeed, once the time dependence is lost, we have $\mathbf{q}_n = 0$ for all $n$, so that equation (6) becomes an identity which cannot be solved for $\mathbf{c}$.

Finally, it should be pointed out that the approach presented in this paper is not limited to models in the form of a single parabolic PDE, such as Eq. (2). It can be applied without significant modification to systems of any number of elliptic, hyperbolic, or elliptic second-order PDEs, as well as higher-order PDEs and ordinary differential equations. In particular, there is no need to separate out the terms such as $\partial_t\mathbf{u}$, which are only present in equations governing temporal evolution. In their absence, the linear system that appears in symbolic regression can be solved using alternative approaches such as singular value decomposition[17].

## Methods

**Experimental system and data collection**. Our experimental setup is the same one as used in ref. [24]. The flow is produced in a shallow electrolyte–dielectric bilayer in a rectangular container, the top view of which is shown in Fig. 1a. The two fluids are immiscible, and both layers have a thickness of 0.3 cm and horizontal extent of $L_x = 17.8$ cm $\times L_y = 22.9$ cm. The container sits in a thermal reservoir, which limits temperature fluctuations to 0. 1 °C, corresponding to a 0.3% bound on working fluid viscosity fluctuations. The liquid dielectric serves as a lubricant to make the flow in the electrolyte layer as close to two-dimensional as possible. However, the no-slip condition at the bottom of the container requires the flow velocity to vary in the vertical direction, regardless of the thickness of the fluid layers; as a result, the fluid flow is not described by a 2D Navier–Stokes equation.

An array of 14 permanent magnets of width $w = 1.27$ cm placed beneath the container generates a magnetic field that is near-sinusoidal in the center of the domain. A direct current with density $\mathbf{J} = J\hat{y}$ passes through the electrolyte layer. Its interaction with the magnetic field produces a Lorentz force $\mathbf{J} \times \mathbf{B}$ that drives the flow. The $z$-component of the magnetic field has been measured at a resolution of ten points per magnet width in each of seven equally spaced horizontal planes throughout the electrolyte layer. The average of these planes is shown in Fig. 4a in comparison with the reconstructed forcing in Fig. 4b. These measurements were only used as a reference to validate the results of our reconstruction procedure.

The electrolyte–dielectric interface is seeded with fluorescent microspheres in order to measure 2D velocity fields quantifying the horizontal flow via PIV[30]. A typical snapshot of the velocity field is shown overlaid on its corresponding vorticity in Fig. 1. The strength of the flow is characterized by the Reynolds number Re$= \bar{u}w/\bar{\nu}$, where $\bar{u}$ is the RMS velocity within the central $8w \times 8w$ region of the domain, and $\bar{\nu} = 3.26 \times 10^{-6}$ m²/s is the characteristic depth-averaged viscosity chosen to allow direct comparison with the results of previous studies of this experimental system[9,24,31–33]. For Re $\lesssim 50$, the vertical ($z$) component of the flow is negligibly small, so that the horizontal flow can be considered divergence-free[9].

Each data set represents the $x$ and $y$ components of the velocity field sampled on a uniform grid ($\Delta x = \Delta y$) within the flow domain and covers a temporal interval of at least 600 s with temporal resolution $\Delta t = 1$ s. The characteristic time scale $\tau$ of the flow varies with Re. At low Re, the flow is periodic, with period of around 120 s. At higher Re, the flow is aperiodic, with autocorrelation time which decreases with Re[31]. The spatial resolution of the data is between 6 and 10 grid points per magnet width $w$, which is the characteristic length scale of the flow. The temporal extent $L_t$ and the spatial resolution of each data set, labeled by the mean Re, are given in Table 1.

**Integration domains and weight functions**. For simplicity, we take the integration domains to be rectangular and centered at different grid points $(x_i, y_i, t_i)$,

$$\Omega_i = \big\{ (x,y,t) \, \big| \, |x - x_i| \le H_x,$$
$$|y - y_i| \le H_y, |t - t_i| \le H_t \big\}, \tag{10}$$

where $H_l$ is the half-width of the integration domain in the direction $l = \{x, y, t\}$. All the domains $\Omega_i$ have the same size, centered spatially and distributed temporally throughout the data set, as shown in Fig. 2. Since integration leads to a reduction of noise due to averaging[17], the domains are chosen to be large in both spatial

## Table 1 Description of the data sets used for the symbolic regression analysis.

| Re | $\tau$ (s) | $\frac{L_t}{\tau}$ | $\frac{2H_x}{L_x}$ | $\frac{2H_y}{L_y}$ | $\frac{2H_t}{L_t}$ | $\frac{\Delta x}{w}$ | $\frac{\Delta t}{\tau}$ |
|---|---|---|---|---|---|---|---|
| 17.88 | 42* | 14 | 0.80 | 0.80 | 0.17 | 0.15 | 0.024 |
| 17.93 | 42* | 14 | 0.80 | 0.80 | 0.17 | 0.15 | 0.024 |
| 19.10 | 42* | 14 | 0.80 | 0.80 | 0.17 | 0.15 | 0.024 |
| 19.75 | 26 | 23 | 0.80 | 0.80 | 0.17 | 0.15 | 0.039 |
| 19.80 | 28 | 21 | 0.80 | 0.80 | 0.17 | 0.15 | 0.036 |
| 22.17 | 28 | 128 | 0.80 | 0.80 | 0.028 | 0.11 | 0.036 |
| 22.27 | 25 | 144 | 0.80 | 0.80 | 0.028 | 0.11 | 0.040 |
| 22.62 | 24 | 150 | 0.80 | 0.80 | 0.028 | 0.11 | 0.042 |
| 23.18 | 26 | 138 | 0.80 | 0.80 | 0.028 | 0.12 | 0.039 |
| 30.88 | 12 | 524 | 0.44 | 0.46 | 0.016 | 0.08 | 0.083 |
| 31.11 | 13 | 866 | 0.48 | 0.50 | 0.0089 | 0.09 | 0.077 |
| 31.26 | 13 | 785 | 0.48 | 0.50 | 0.0098 | 0.09 | 0.077 |
| 35.52 | 9 | 2001 | 0.48 | 0.50 | 0.0056 | 0.09 | 0.111 |
| 35.67 | 9 | 2001 | 0.48 | 0.50 | 0.0056 | 0.09 | 0.111 |
| 36.34 | 8 | 149 | 0.81 | 0.85 | 0.083 | 0.09 | 0.125 |

Re denotes the mean Reynolds number. Times $\tau$ marked with an asterisk (*) represent temporal period, whereas those without represent autocorrelation time.

directions. Their spatial width $2H_x \times 2H_y$ was chosen to be slightly smaller than the size $L_x \times L_y$ of the flow domain to avoid the regions near the side walls where PIV is noisier than in the bulk. The temporal width $2H_t$ was chosen to be smaller than the temporal extent $L_t$ of the data set to limit overlap between different integration domains, so that rows of equation (6) could remain linearly independent. Specific values of $H_x$, $H_y$, and $H_t$ for each data set are given in Table 1.

As mentioned previously, each partial derivative of the velocity field increases the noise that is inevitably present in the PIV data. Hence, the derivatives are transferred onto the smooth, noiseless weight functions $\mathbf{w}_j$ whenever possible. Consider for illustration the term $F_0 = \partial_t\mathbf{u}$. Using integration by parts we obtain

$$\langle \mathbf{w}_j, \partial_t\mathbf{u} \rangle_i = -\langle \partial_t\mathbf{w}_j, \mathbf{u} \rangle_i, \tag{11}$$

if the boundary terms are eliminated by requiring $\mathbf{w}_j = 0$ at $t = t_i \pm H_t$. The complete set of boundary conditions[18] requires that $\mathbf{w}_j$ and its spatial derivatives up to second-order vanish at the boundary of the integration domain. Some nonlinear terms in Eq. (2), such as $\omega^2\mathbf{u}$, do not allow all derivatives to be transferred onto $\mathbf{w}_j$ via integration by parts. In such cases, the remaining derivatives on $\mathbf{u}$ are computed in Fourier space utilizing both a Tukey-like windowing function and a low-pass filter.

Furthermore, the weight functions should be chosen such that the integrals involving the latent fields disappear. To remove the dependence on the time-independent forcing term, we require that $\mathbf{w}_j$ be an odd function in time, such that

$$\int_{-H_t}^{H_t} \mathbf{w}_j \, dt = 0, \tag{12}$$

We also constrain our weight function to the form

$$\mathbf{w}_j = \nabla \times [\hat{z}\phi_j(x,y,t)], \tag{13}$$

so that

$$\langle \mathbf{w}_j, \nabla p \rangle_i = -\langle \nabla \cdot \mathbf{w}_j, p \rangle_i = 0, \tag{14}$$

eliminating the dependence on pressure.

All of the above constraints can be satisfied by choosing the scalar fields $\phi_j$ in the form

$$\phi_j(x,y,t) = P_\lambda(x')P_\mu(y')P_\nu(t')E_\alpha(x')E_\beta(y')E_\gamma(t'), \tag{15}$$

where $P_m(\cdot)$ is a Legendre polynomial,

$$E_\alpha(w) = (1 - w^2)^\alpha, \tag{16}$$

is an envelope function, and the prime denotes coordinates scaled by the integration domain size: $x' = (x - x_i)/H_x$, $y' = (y - y_i)/H_y$, $t' = (t - t')/H_t$. Each integral over $\Omega_i$ is evaluated numerically using the trapezoidal rule, with the accuracy of the numerical quadrature controlled by the integers $\alpha$, $\beta$, and $\gamma$[17]. Here we set $\alpha = \beta = \gamma = 6$ to allow the use of PIV data that is relatively sparse. For reference, regression based on direct evaluation of derivatives via a polynomial method[11] requires about 20 grid points per magnet width (e.g., 2–3 times higher than in our data sets).

Unlike ref. [11] which considered symbolic regression for synthetic data, multiple weight functions labeled by integer indices $j = \{\lambda, \mu, \nu\}$ were used here to sample the data more thoroughly, while keeping the large integration domains from overlapping too much for the shorter data sets. The constraint (12) requires $\nu$ to be an odd integer. Here we used all combinations of $\lambda$ and $\mu$ set to either 0 or 1 and

$v = 1$, i.e., a total of four weight functions for each integration domain (this number could be increased further to improve the model reconstruction accuracy). The total number of equations in the system defined by equation (6) is therefore $K = 4I$, where $I$ is the total number of integration domains. The system has to be overdetermined, $K > N$; we chose $I = 50$ which satisfies this condition. A higher value would further increase the accuracy and robustness of the method.

**Reconstructing the pressure and forcing field.** Once the parsimonious model describing a particular data set has been found, the horizontal forcing profile $\mathbf{f}(\mathbf{x})$ and pressure $p(\mathbf{x},t)$ can be computed using the Helmholtz decomposition of the vector field $\mathbf{s}(\mathbf{x},t)$ in Eq. (4). Specifically,

$$p(\mathbf{x}, t) = -\rho \iint \frac{i\mathbf{k} \cdot \hat{\mathbf{s}}(\mathbf{k}, t)}{\mathbf{k} \cdot \mathbf{k}} e^{-i\mathbf{k} \cdot \mathbf{x}} d\mathbf{k} \tag{17}$$

and

$$\mathbf{f}(\mathbf{x}, t) = -\rho \iint \frac{\mathbf{k} \times [\mathbf{k} \times \hat{\mathbf{s}}(\mathbf{k}, t)]}{\mathbf{k} \cdot \mathbf{k}} e^{-i\mathbf{k} \cdot \mathbf{x}} d\mathbf{k}, \tag{18}$$

where

$$\hat{\mathbf{s}}(\mathbf{k}, t) = \hat{F}_0(\mathbf{k}, t) - \sum_{n=1}^{7} c_n \hat{\mathbf{F}}_n(\mathbf{k}, t). \tag{19}$$

and

$$\hat{\mathbf{F}}_n(\mathbf{k}, t) = \frac{1}{(2\pi)^2} \iint \mathbf{F}_n(\mathbf{x}, t) e^{i\mathbf{k} \cdot \mathbf{x}} d\mathbf{x}. \tag{20}$$

The latent fields are reconstructed without the benefit of the weak formulation, which plays a crucial role in increasing the robustness of symbolic regression in the presence of noise. Since some of the terms $\mathbf{F}_n(\mathbf{x}, t)$ involve derivatives, which amplify noise, the respective Fourier transforms $\hat{F}_n(\mathbf{k}, t)$ are low-pass filtered by eliminating frequencies $|k_x| > 2k_0$ and $|k_y| > 2k_0$ where $k_0 = \pi/w$ is the wavenumber corresponding to the wavelength $2w$ of the magnet array. This cut-off frequency is chosen empirically to balance the inclusion of relevant modes and the exclusion of modes corrupted by noise. The spatial derivatives were computed spectrally and the temporal derivative term was computed using a second-order central difference.

Note that $\mathbf{f} = \rho \nabla \times A$ involves an extra derivative compared with $p = \rho\phi$, which decreases its accuracy for noisy data. Since $\mathbf{f}$ is stationary in our experiment, its accuracy can be improved substantially by temporally averaging Eq. (18).

## Data availability
Data sets containing velocity fields and their gradients, as well as the source data used to construct Figure 3 are available from the Open Science Framework at https://doi.org/10.17605/osf.io/tez6c.

## Code availability
MATLAB codes used to identify the governing equations can be found on GitHub at https://doi.org/10.5281/zenodo.4653308. Any other requests should be made to the corresponding author.

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

## Acknowledgements

This material is based upon work supported by NSF under Grant Nos. CMMI-1725587 and CMMI-2028454. The experimental data used in this work were produced by Jeff Tithof. The magnetic field measurements were performed with assistance from Charles Haynes.

## Author contributions

P.A.K.R. was responsible for conducting data analysis and interpretation of the results. L.M.K. was responsible for performing fluid flow experiments, data acquisition, and PIV analysis. M.F.S. was responsible for experimental design. R.O.G. was responsible for concept and research design. All authors were involved in the preparation of the manuscript, read and approved the final version.

## Competing interests
The authors declare no competing interests.
