## [Peer Review File · Nature Communications]

Reviewer #1 (Remarks to the Author):

This paper demonstrates how techniques developed by the same authors in previous work (Reinbold, Gurevich, Grigoriev, *Phys. Rev. E*, 2020) can be combined with domain knowledge to reveal the underlying governing equations of a complicated spatiotemporal system from real experimental data.

These techniques were developed to address two main issues with existing methods (Schaeffer, *Proc. Royal Soc. A*, 2017) and (Rudy, Brunton, Proctor, Kutz, *Sci. Advances*, 2017) that have prevented their use on experimental data: 1. sensitivity to noise caused by numerical differentiation and 2. existence of unobserved latent variables or fields in the models.

The key innovation is the use of a weak formulation in order to pass otherwise noise-amplifying differential operators onto smooth test functions.

Careful choice of test functions together with domain knowledge allows them to cancel the unknown fields from the learning problem.

Yet until the present work, the techniques (both those of the present authors as well as Schaeffer and Rudy et al.) remained un-demonstrated in the real-world setting, only having been tested on simulated model problems.

This successful demonstration of these advanced learning methods is a major contribution and in my opinion merits publication. I do have the following comments:

1. While I think the work is strong as-is, it could be made stronger if the approach was compared with that of (Schaeffer, *Proc. Royal Soc. A*, 2017) and (Rudy, Brunton, Proctor, Kutz, *Sci. Advances*, 2017). This may not be straight-forward because of the unknown pressure field. It might be done by enforcement of the divergence-free condition which gives the pressure as a solution of a Poisson equation with unknown coefficients of the right hand side. A Greens function solution for pressure could be substituted back into equation (2) to eliminate it. Mainly, it would be great to see if/how the finite-difference-based noise amplification deteriorates the solution compared to the weak formulation. Of course this would be a lot of work and I am certainly not insisting that you do it.

2. The method requires choosing weight functions so that the integrals with latent fields disappear. Is this always possible in general, or just in special cases? When is it possible to find such weight functions and how difficult is it when the dictionary is large. How much domain knowledge is needed to properly choose the weight functions? A few words about these issues might be helpful.

3. I think the use of the relative residual eq.7 is an interesting idea and I see that it comes from (Reinbold and Grigoriev, *Phys. Rev. E*, 2019). Is this metric a new invention or has it been used elsewhere? Have you considered Akaike and Bayes information criteria (AIC) and (BIC)? These are methods that I think most readers will be more familiar with for model selection and it might be worth saying a few words about them.

4. Model uncertainty is quantified using an ensemble of integration domains. It's not obvious to me how this addresses the issue of quantities like c_6 because variations in c_6 will always be in the null-space of Q regardless of how the domains are selected. Least squares or sparse regression should always return $c_6=0$, but really the true value of c_6 simply cannot be inferred from the data either because of a true physical constraint like incompressibility, or because the only available velocity field data happened to have zero divergence. While this is not an issue here, I can imagine this being a serious problem if there is a conserved quantity like energy. If your data is obtained at a single value of the conserved quantity, you won't be able to determine the corresponding coefficients. I think it would be more correct to say that combinations of quantities in the null space of all Q in the ensemble are fundamentally un-learnable given the available data in the absence of additional domain knowledge. These unknowable quantities are actually very useful information for a user of the method to have as it could be used to design additional experiments to make Q full column rank.

5. Robustness of l_1 minimization approaches like SINDy are studied by H. Schaeffer, G. Tran, and R. Ward "Extracting Sparse High-Dimensional Dynamics from Limited Data", *SIAM J. Appl. Math.*, 2018. This paper might be worth citing. This paper also deals with what kind of data one needs in order to identify parameters in dynamical systems. In particular, identification is possible when the

system is ergodic (explores the state space), but many different initial conditions are needed when the dynamics are not ergodic. This may help explain why the models become less accurate at lower Reynolds number.

6. In your method, you use the weight functions (test functions) to eliminate dependence on fields at the boundary. Suppose I have a spatiotemporal PDE whose boundary conditions I do not know, but I want to find out. Can the boundary conditions also be learned from data? I imagine that these terms pop out of the integration by parts over domains that include the boundary and can be put into essentially the same framework together with additional expansions for the boundary conditions in some dictionary. Is this true? It might be worth a few words, but mostly I'm just curious.

7. At present, the use of sparse regression is limited to linear combinations of symbolic terms. Do you think one could use your approach in conjunction with genetic programming-based symbolic regression where the symbolic terms are allowed to be composed in more complicated ways?

8. Finally, in equation (2), what is u^2 ?

Thanks for sharing this paper. It is wonderful to see a sophisticated data-driven method for model identification successfully demonstrated in a physical experiment! In my experience this is a rare thing.

Best,
Sam Otto
Princeton University, Dept. Mechanical and Aerospace Engineering

Reviewer #3 (Remarks to the Author):

In this article the authors:

1. propose combining a data-driven methodology with some general physical principles to enable the discovery of a model of a spatio-temporal system from data.
2. illustrate this using an experimental weakly turbulent fluid flow
3. show that this hybrid approach allows reconstruction of the inaccessible variables – the pressure and forcing field driving the flow.

The statistical analysis is clear, sufficient details are provided and the paper is well written, and hence should be reproducible by an independent researcher.

While the methodology is novel, it contains ideas that overlap with Brunton et al. (PNAS, 2016, <https://doi.org/10.1073/pnas.1517384113>) and related works. Given the relative simplicity and specificity of the example on which the method is illustrated, it is unclear to me that this will be of interest to the wider field. In particular, even within the broad and complex discipline of fluid dynamics, it is unclear if this method will hold up in more complex turbulent systems and 3 dimensions (in space).

Hence, I am not particularly convinced, given the rather simplified fluid flow experiments on which the results are shown, that this method will actually work well for non-equilibrium spatially-extended systems from high-dimensional data that is both noisy and incomplete.

Given that the example shown has limited scope, I do not feel strongly that the paper will influence thinking in the field, although it does seem like a potentially powerful approach.

My major concerns are:

1. The method is not benchmarked against the leading competitors in the field (eg. sparse identification of nonlinear dynamical systems, and other machine learning methods).
2. The method is not shown to work for a wide range of PDEs, and for more turbulent fluid flows in 3D.

2. The method is not tested on a more complex system where the governing equations are only partially known or unknown, where physical intuition becomes truly critical.

Further evidence that would be required to strengthen the conclusions are:

1. showing that the method works on a 3D turbulent flow.
2. showing that the method works on a system governed by another PDE or set of PDEs.
3. a more complex system (biological, for example) where the underlying governing equations are unknown but there exists sufficient knowledge in the field to judge the validity of the model discovered.

Given my major concerns, this manuscript is not of sufficiently high impact for publication in this journal. I would be willing to review a revised manuscript that carefully addresses my concerns above and presents new research to strengthen the conclusions.

Referee 1

We would like to thank the referee for the time and effort devoted to reviewing our manuscript. Below, we have listed the comments from the referee (in black) and our responses (in blue).

1. While I think the work is strong as-is, it could be made stronger if the approach was compared with that of (Schaeffer, Proc. Royal Soc. A, 2017) and (Rudy, Brunton, Proctor, Kutz, Sci. Advances, 2017). This may not be straight-forward because of the unknown pressure field. It might be done by enforcement of the divergence-free condition which gives the pressure as a solution of a Poisson equation with unknown coefficients of the right hand side. A Greens function solution for pressure could be substituted back into equation (2) to eliminate it. Mainly, it would be great to see if/how the finite-difference-based noise amplification deteriorates the solution compared to the weak formulation. Of course this would be a lot of work and I am certainly not insisting that you do it.

The referee is absolutely correct that pressure could be computed from velocity field via the appropriate Green's function for the Navier-Stokes equation with no forcing or with known forcing. In the present problem, however, the forcing is not known and it is generally not divergence-free either, so the pressure cannot be computed without the knowledge of the forcing, period. Even if the forcing were known, the pressure-Poisson equation would contain terms of up to third derivative in space, leading to extremely noisy results. Furthermore, the solution to the pressure-Poisson equation in terms of the Green's function involves boundary terms. PIV measurements are particularly noisy for slow flows near no-slip boundaries, so evaluation of these boundary terms will add even more noise to the pressure.

We added a short statement to this account and another one pointing out that a comparison of the weak formulation with the strong formulation for the present problem has already been made in our earlier paper (Ref. [16]) which used synthetic data to show that strong formulation falls apart at very small levels of noise.

2. The method requires choosing weight functions so that the integrals with latent fields disappear. Is this always possible in general, or just in special cases? When is it possible to find such weight functions and how difficult is it when the dictionary is large. How much domain knowledge is needed to properly choose the weight functions? A few words about these issues might be helpful.

No approach is truly universal and each approach has limitations. In the present case, the choice of the weight functions is based on both the form of the terms involving latent fields (the pressure field entering via the gradient) and some domain knowledge (that forcing is time-independent). This illustrates that the success of our approach is ultimately due to its hybrid nature combining first-principles or domain knowledge with the data-driven methods. For example, in the absence of any information about the forcing field, we would not be able to choose the weight functions to eliminate the dependence on the forcing field. We added a paragraph addressing these issues at the end of the paper.

The choice of the weight function(s) does not depend on the total number of terms in the candidate model. It does depend on the number of terms involving latent variables and their functional form, however. Essentially, every term in the candidate model involving latent variables imposes an additional constraint on the weight function(s), as our example illustrates. If there are too many terms involving latent variables, it may not be possible to satisfy all the constraints, and the presented approach will break down, indicating the limitations of what can be achieved with incomplete information.

3. I think the use of the relative residual eq.7 is an interesting idea and I see that it comes from (Reinbold and Grigoriev, Phys. Rev. E, 2019). Is this metric a new invention or has it been used elsewhere? Have you considered Akaike and Bayes information criteria (AIC) and (BIC)? These are methods that I think most readers will be more familiar with for model selection and it might be worth saying a few words about them.

The use of a relative residual is certainly not new, but it is worth pointing out its direct comparison to AIC and BIC isn't quite apples-to-apples. The relative residual measures how well the model fits the data, while AIC and BIC also encode (via their penalization of) model complexity. It would be more appropriate to compare sequential thresholding with model selection via minimizing AIC or BIC. A comparison between these approaches in the context of ODEs has already been performed by Mangan et al. (Proc. Roy. Soc. 2017). That study found that the best model identified by SINDy was the only one to have strong support, while the rest had no support, in the AIC sense. We added a reference to this paper, but a more detailed comparison with information-theoretic methods is outside the scope of our work.

4. Model uncertainty is quantified using an ensemble of integration domains. It's not obvious to me how this addresses the issue of quantities like c_6 because variations in c_6 will always be in the null-space of Q regardless of how the domains are selected. Least squares or sparse regression should always return $c_6=0$, but really the true value of c_6 simply cannot be inferred from the data either because of a true physical constraint like incompressibility, or because the only available velocity field data happened to have zero divergence. While this is not an issue here, I can imagine this being a serious problem if there is a conserved quantity like energy. If your data is obtained at a single value of the conserved quantity, you won't be able to determine the corresponding coefficients. I think it would be more correct to say that combinations of quantities in the null space of all Q in the ensemble are fundamentally un-learnable given the available data in the absence of additional domain knowledge. These unknowable quantities are actually very useful information for a user of the method to have as it could be used to design additional experiments to make Q full column rank.

We completely agree. In fact, we have addressed this precise issue previously in Ref. [10]. There are numerous circumstances in which some terms in the candidate model could either vanish or be too small for the respective coefficient to be recovered. Our response to referee 3 provides another good example, where the viscous term in the Navier-Stokes equation may be extremely small. Such terms will generally be eliminated from the model by symbolic regression. This, however, is not a limitation of the

approach, since the resulting model accurately reproduces the data set from which it was identified and hence is a good model by definition. This does not mean it is the most general model, however.

Another example of this is the breakdown of our approach for steady flows, when all of the terms are in the null space, as mentioned in the paper. In the end, the choice of excluding or including some terms is truly a philosophical question. For instance, should external forcing be included in the Navier-Stokes equation? That depends on whether the model aims to describe fluid flows that are forced or unforced. The key is to acknowledge the role of the data: if a more general form of a model is sought, the data should be chosen accordingly (e.g., data describing different energies for energy-conserving systems). This issue is far too general to include a discussion of in the present paper without diluting its focus.

5. Robustness of L1 minimization approaches like SINDy are studied by H. Schaeffer, G. Tran, and R. Ward “Extracting Sparse High-Dimensional Dynamics from Limited Data”, SIAM J. Appl. Math., 2018. This paper might be worth citing. This paper also deals with what kind of data one needs in order to identify parameters in dynamical systems. In particular, identification is possible when the system is ergodic (explores the state space), but many different initial conditions are needed when the dynamics are not ergodic. This may help explain why the models become less accurate at lower Reynolds number.

We agree and cited the paper.

6. In your method, you use the weight functions (test functions) to eliminate dependence on fields at the boundary. Suppose I have a spatiotemporal PDE whose boundary conditions I do not know, but I want to find out. Can the boundary conditions also be learned from data? I imagine that these terms pop out of the integration by parts over domains that include the boundary and can be put into essentially the same framework together with additional expansions for the boundary conditions in some dictionary. Is this true? It might be worth a few words, but mostly I’m just curious.

Our formalism can certainly be extended to learning the physical boundary conditions. In this case, one should simply use integrals over the corresponding surface (and time), rather than volume (and time). It would certainly be interesting to validate this idea in future work.

7. At present, the use of sparse regression is limited to linear combinations of symbolic terms. Do you think one could use your approach in conjunction with genetic programming-based symbolic regression where the symbolic terms are allowed to be composed in more complicated ways?

Whether our approach extends to nonlinear regression is unclear, regardless of whether genetic programming is used. For instance, it is unclear whether the weak formulation of a governing equation with complex nonlinearities can be used to either eliminate the dependence on latent variables or reduce the error in evaluating high order derivatives

via integration by parts. Furthermore, it is far less trivial, albeit not impossible, to ensure that the tensor structure is preserved.

8. Finally, in equation (2), what is u^2 ?

$u^2 = \mathbf{u} \cdot \mathbf{u}$ is the squared magnitude of the velocity. We updated the notations in the paper to clarify this.

Referee 3

We would like to thank the referee for the time and effort devoted to reviewing our manuscript. Below, we have listed the comments from the referee (in black) and our responses (in blue).

1. The method is not benchmarked against the leading competitors in the field (eg. sparse identification of nonlinear dynamical systems, and other machine learning methods).

This is not the case. Even though this particular paper does not contain such benchmarking, the papers on which it is based (Refs. [15] and [16]) provide a detailed comparison of the weak formulation with the more traditional approaches, such as SINDy, based on the strong formulation. These papers show that our approach is far superior to those more traditional approaches in its accuracy, robustness, and versatility. To make this more explicit in the manuscript, we added a short statement in paragraph 1 of the subsection "Weak formulation of the model".

Direct comparison with the more traditional approaches for the present problem is simply not possible, since those approaches cannot cope with incomplete data, as we explain in response to comment 1 of referee 1. Please also note that no previous method has been demonstrated to work for experimental data, whether complete or not. The main goal of the present paper is not to repeat the analysis presented in Refs. [15] and [16], but rather to demonstrate that our approach works well for experimental data.

2. The method is not shown to work for a wide range of PDEs, and for more turbulent fluid flows in 3D.

We respectfully disagree. Our approach has been demonstrated to work for a representative range of PDEs. In particular, Ref. [16] has demonstrated that it works remarkably well for every PDE (or system of PDEs) for which more traditional approaches struggle, i.e., the Kuramoto-Sivashinsky equation, the lambda-omega model, and the Navier-Stokes equation in 2D.

Our approach is applicable to data in any number of dimensions, including three or more, and any kind of data, chaotic or not. In fact, it performs the best for data that is complicated and has a lot of variability. To illustrate this, we used highly turbulent channel flow data available at the JHU database. The data set contains fully-resolved numerical solution of the 3D Navier-Stokes equation

$$\partial_t \mathbf{u} + (\mathbf{u} \cdot \nabla) \mathbf{u} = -\nabla p + Re^{-1} \nabla^2 \mathbf{u}, \quad (1)$$

with $Re = 20,000$.

To illustrate our approach, we used a candidate model in the form

$$\partial_t \mathbf{u} = c_1 (\mathbf{u} \cdot \nabla) \mathbf{u} + c_2 \nabla^2 \mathbf{u} + c_3 \mathbf{u} + c_4 u^2 \mathbf{u} + c_5 \nabla p, \quad (2)$$

which can be obtained with the help of the same arguments as used in the 2D case. Even though the database contains the pressure information, we discard it for consistency with the approach described in our paper. This requires using a weight function

	Bulk	Boundary
c_1	-1.00026	-1.00035
c_2	0	4.9895×10^{-5}
c_3	0	0
c_4	0	0
H_x	0.333	0.333
H_y	0.200	2×10^{-2}
H_z	0.200	0.200
H_t	0.250	0.250
n_x	27	27
n_y	33	34
n_z	32	32
n_t	39	39
N_d	50	50
η	5.4025×10^{-3}	3.9852×10^{-3}

Table 1: Identified values of the coefficients and the hyperparameters used for the turbulent 3D channel flow. The size of the integration domains in non-dimensional units are given by H_i , the number of grid points this corresponds to is given by n_i , and N_d denotes the total number of integration domains used to construct the linear system. The “bulk” refers to integration domains located in the center of the channel, the “boundary” refers to integration domains confined to either of the two boundary layers. In both cases the integration domains were randomly distributed in time, streamwise, and spanwise direction.

that has the same form as in the 2D case, $\mathbf{w}(x, y, z, t) = \nabla \times \phi$. For simplicity, we chose $\phi = [0, 1, 1]\phi(x, y, z, t)$, where ϕ is the product of one-dimensional envelope functions defined in the main text, this time including one in the z -direction as well.

As expected, depending on where the data is taken from, our approach identifies either the Navier-Stokes equation or the Euler equation. For data from the bulk (away from the wall), the Euler equation is recovered. For data from the boundary layer next to a no-slip wall, the Navier-Stokes equation is recovered. This is in perfect agreement with our understanding of high-Re turbulent flows: viscosity is important in the boundary (viscous) layer, but its effect becomes negligible further away, in the log-layer and beyond. Table 1 summarizes the relevant information about various parameters used, and the results obtained, in each of the two cases. Note that the coefficients c_n were identified with exceptional accuracy. The highest error (about one percent) corresponds to $c_2 = Re^{-1}$ and is due to the discretization errors of the DNS.

3. The method is not tested on a more complex system where the governing equations are only partially known or unknown, where physical intuition becomes truly critical.

We respectfully disagree. This comment certainly describes every single study that precedes our work in the context of PDE models. All of those studies used synthetic

data generated using a known – mostly rather simple – model to recover the same model. This paper is the only exception, which does exactly what the referee is asking for. Specifically, experimental data is used to identify a model, and this model is then compared with other models that have been derived using first principles approaches – or guessed using physical intuition – for the same physical system. Note that it is not known what the correct model of weakly turbulent fluid flow in a thin layer (or bi-layer) of fluid is. In fact, even “derived” models rely on unproven – and in some regimes incorrect – assumptions and provide at best an approximate description of experiment as shown in Ref. [22].

4. Further evidence that would be required to strengthen the conclusions are:
 - (a) showing that the method works on a 3D turbulent flow
 - (b) showing that the method works on a system governed by another PDE or set of PDEs.
 - (c) a more complex system (biological, for example) where the underlying governing equations are unknown but there exists sufficient knowledge in the field to judge the validity of the model discovered.

All of this is addressed by our responses to comments 2 and 3 above.

Reviewer #1 (Remarks to the Author):

All of my comments and concerns have been thoughtfully addressed. This work shows how domain knowledge can be incorporated into a weak formulation of the learning problem for governing equations in order to overcome two well-known issues with existing techniques. Namely, that all state variables (spatial fields) have to be known and that differentiation in the strong sense amplifies noise in the data to such a degree that it limits practical application. The present work demonstrates that the proposed method overcomes these limitations on a challenging experimental data set. This is significant since the vast majority of prior work relies on simulated data. The lack of direct comparisons with other methods is not a deficiency of the paper, but a reflection of the fact that there are no other comparable methods that are capable of solving the problem at hand. Therefore, I recommend this paper for publication and I am excited to see what else this class of methods is capable of!

Samuel Otto

Dept. of Mechanical and Aerospace Engineering, Princeton University